# Comfort Prediction of Office Chair Surface Material Based on the ISSA-LSSVM

**DOI:** 10.3390/s22249822

**Published:** 2022-12-14

**Authors:** Xuyinglong Zhang, Zilian Cheng, Mengyang Zhang, Xiaodong Zhu, Xianquan Zhang

**Affiliations:** 1Material Science and Engineering College, Northeast Forestry University, Harbin 15000, China; 2Department of Performing Arts and Culture, The Catholic University, Bucheon 14662, Republic of Korea; 3Engineering and Technology College, Northeast Forestry University, Harbin 15000, China

**Keywords:** office chair, improved sparrow search algorithm, least square support vector machine, comfort, prediction

## Abstract

This study serves the purpose of assisting users in selecting a comfortable seat surface material for office chairs and enhancing users’ comfort while using office chairs. To address the issue that the selection of traditional seat surface material is too subjective and that the prediction effect is poor, an improved sparrow search algorithm (ISSA) optimized least squares support vector machine (LSSVM) method for office chair seat surface material comfort prediction has been proposed. Sparrow Search Algorithm (SSA) was optimized with Sobol sequences, nonlinear inertial weights, and a crisscross optimization algorithm to produce the Improved Sparrow Search Algorithm (ISSA), and then the relevant parameters of the LSSVM algorithm were optimized with the modified algorithm to improve its prediction performance. The prediction accuracy of the ISSA-LSSVM model is as high as 95.75% by combining the body pressure distribution experiments; the root mean square error (RMSE) is 0.29; the goodness of fit (R^2^) is 0.92; the mean absolute error (MAE) is 0.24; the standard deviation (RSD) is 5.99%. The ISSA-LSSVM model predicts seat surface material comfort more accurately and reliably. This strategy can assist consumers to narrow down their seat surface material choices and even suggest an optimal selection. In this way, it can boost users’ pleasure with office chairs, which has great potential for wide application.

## 1. Introduction

With the rise of industrial automation and the popularity of artificial intelligence technologies, an increasing number of people are engaged in sedentary labor. In industrialized nations such as Europe and the United States, around 75% of workers are involved in sitting-related employment and 54% of working time is spent in a chair [1]. Workplace seating comfort exerts a great impact on people’s daily life.

Mastrigt et al. [2] reviewed seat comfort research and found that comfort can be communicated directly through the user’s subjective feel, and that human comfort is not only directly related to the seat’s attributes but also the external surroundings. Irshad et al. [3] used objective methods including heart rate and eye movements to assess the thermal comfort of sleepers under thermoelectric air duct cooling system (TE-DA) and natural ventilation circumstances. They concluded that TE-DA conditions provided greater thermal comfort and sleep quality. Irshad et al. [4] conducted additional research based on the initial study by examining the sleep comfort of a thermoelectric air duct cooling system (TE-DA) with varying operating powers. Using objective indications such as measuring the user’s skin temperature and heart rate in conjunction with subjective comfort analyses, the study found that increasing the power to 840 W significantly decreased the user’s sleep experience. Therefore, the environment has a substantial impact on comfort and must be controlled as a key variable prior to undertaking specific investigations.

Currently, the methods for analyzing seat comfort are predominately based on the development of applied mathematical models. In addition, subjective evaluation methods are predominantly employed, such as the general comfort scale offered by Helander et al. and Shackle et al. [5], the local discomfort scale provided by Corlett and Bishop, and the multidimensional seat evaluation scale proposed by Zhang et al. Based on a study by Helander et al. [5], Ziolek et al. [6] used the Subjective Comfort and Discomfort Scale to evaluate the comfort of vehicle seats for long-distance driving and discovered that subjective driver comfort varied significantly between seat types. Levart et al. [7] established an evaluation model for car seat comfort to compare the discomfort of various types of car seats, using the more subjective fuzzy mathematical theory. In fact, the majority of conventional mathematical modeling techniques, such as fuzzy mathematics, quantify them on the basis of subjective evaluations, which serve a similar function as subjective scales. Although subjective evaluation techniques play an essential part in comfort analysis, the outcomes of such research are questionable due to their excessive reliance on user perceptions and lack of objectivity.

Zemp et al. [8] examined the body pressure distribution test as a dependable approach for evaluating seat comfort, among several objective metrics. Li et al. [9] demonstrated a substantial correlation between the subjective comfort rating of the seat and the mean pressure mean peak pressure of the body pressure distribution test measurements. Young et al. [10] discovered that the buttocks are one of the most influential factors in the comfort of a seat. As the seat surface is the primary contact between the person and the seat, it is essential to enhance the seat surface material’s comfort level. Kumar [11] and others have proven that pressure between the individual and the seat contact surface can be reduced and pressure sores prevented by using adequate seat materials Moreover, seat comfort could be improved by optimizing seat material qualities.

Experts and researchers have been focusing on how to successfully integrate the subjective assessment approach with the objective evaluation method to produce a more scientific comfort rating system. Some researchers have attempted to construct a comfort evaluation system by combining machine learning algorithms with subjective evaluation methods. Lerspalungsanti et al. [12] showed that artificial neural network algorithms could predict users’ subjective comfort with more accuracy than linear regression. Deploying artificial neural network techniques, Zhao et al. [13] analyzed the comfort of civil airliners with varied seat pitches in order to forecast the comfort of aircraft seats. Li et al. [14] used objective evaluation indicators such as pressure distribution parameters like input variables and subjective evaluation indicators like output indicators to establish a comfort evaluation model, with the use of the neural network method. The discussion of the applicability of the BP neural network algorithm in evaluating and predicting the differences in comfort between various types of wheelchairs for the elderly followed. Long Jiang et al. [15] proposed employing a heuristic algorithm-artificial bee colony method to optimize BP neural network-related parameters, enhance the performance of the model, and forecast the comfort of automobile seats. Compared to the conventional BP neural network model, the improved prediction model was more stable and dependable, with higher accuracy of its predictions. Even though the performance of conventional machine learning and deep learning algorithms has been enhanced by optimization through heuristic algorithms such as the genetic algorithm, ant colony algorithm, and artificial bee colony algorithm, there are still certain issues. The artificial bee colony algorithm is insufficient in local searching ability, the algorithm iteration period is long, and it cannot balance exploration and development ability. The ant colony algorithm has more control parameters and is correlated, so performance improvement must rely on continuous trial and error and the following adjustment. The genetic algorithm coding process is more complex and fraught with issues.

The majority of the current research on seating comfort focuses on vehicles and the aerospace industry, with less research on office chairs; the same is true for the effect of the seat’s structure on comfort, leaving seat surface material comfort inadequately researched. Office chair seat material is the direct carrier between the user and the seat, thus its comfort must be thoroughly researched and evaluated. The least squares support vector machine (LSSVM) is a prediction model with strong structural performance that overcomes anomalous regressions resulting from coarse and volatile data sets [16]. The present investigation demonstrates that the sparrow search algorithm [17] has the benefits of a simple structure, simple implementation, few control parameters, and great local search capacity when compared to the conventional heuristic method. However, the approach is prone to local optimality, lacks the capacity to find the global optimal solution, and has issues such as low convergence accuracy. The goal of this work is to develop a robust and accurate model for predicting the surface comfort of office chairs. Firstly, according to the inherent flaws of the sparrow method, the starting population is mapped with the Sobol sequence so that it is more equally distributed. Then, non-linear inertia weights were employed to enhance the algorithm’s global search capability and convergence efficiency. The vertical and horizontal crossover method was then implemented to balance the algorithm’s local search ability and global exploitation capacity and to prevent it from sliding into a local optimum. Afterwards, the improved sparrow search algorithm (ISSA) was utilized to overcome the difficulty to discover the best parameters of the LSSVM model fast and precisely. Finally, the ISSA-LSSVM model was utilized to integrate the user’s subjective comfort evaluation with the body pressure distribution test to produce a more scientific and realistic comfort prediction model.

## 2. Materials and Methods

### 2.1. Least Squares Support Vector Machine

The LSSVM method is an extension of the conventional support vector machine (SVM) that can effectively deal with small-sample nonlinear situations [18]. By solving a system of linear equations [19], the problem of convex quadratic programming is converted to SVM in order to lower the solution difficulty and optimize the overall processing performance. The following are the stages [20,21]:

The LSSVM regression estimation function is:(1)$$\begin{array}{c} f\left(x\right)=W^{T}\varphi \left(x\right)+a \end{array}$$
where $W^{T}$ represents the hyperplane weight coefficient, $\varphi \left(x\right)$ represents the mapping function, and a represents the offset.

In accordance with the principle of structural risk reduction, LSSVM defines the objective function J and constraints of the optimization problem as follows:(2)$$\begin{array}{c} \left\{\begin{array}{c} minJ\left(w,e\right)=\frac{1}{2}w^{T}w+\frac{1}{2}r\overset{n}{\underset{i=1}{\sum }}e_{i}^{2}\\ s.t.y_{i}=w^{T}\varphi \left(x_{i}\right)+b+e_{i} \end{array}\right. \end{array}$$
where r represents the penalty coefficient, $e_{i}$ represents the error, and *b* represents the threshold.

Introduce Lagrange multipliers such that $\alpha_{i}\geq 0$ and construct Lagrange functions before carrying out the relevant transformations.
(3)$$\begin{array}{c} L\left(w,b,e,\alpha \right)=\frac{1}{2}w^{2}+\frac{1}{2}r\overset{n}{\underset{i=1}{\sum }}e_{i}^{2}-\overset{n}{\underset{i=1}{\sum }}\alpha_{i}\left(w^{T}\varphi \left(x_{i}\right)+b+e_{i}-y_{i}\right) \end{array}$$

The partial derivatives of Equation (3) are determined independently, so that the derivative equals zero. To eliminate w and $e_{i}$, the following system of linear equations can be produced.
(4)$$\begin{array}{c} \left|\begin{array}{cc} 0 & E^{T}\\ E & \Omega +r^{-1}I \end{array}\right|\left|\begin{array}{c} b\\ \alpha \end{array}\right|=\left|\begin{array}{c} 0\\ y \end{array}\right| \end{array}$$
where I is the unit matrix; Ω is the kernel mapping matrix.

In this study, the radial basis function (RBF) was used. Compared with other kernel functions, the RBF kernel function is capable of nonlinear mapping with fewer computing resources. The expression is as follows:(5)$$\begin{array}{c} K\left(x,x_{i}\right)=exp\left(-\frac{1}{2\sigma^{2}}x-x_{i}{}_{2}^{2}\right) \end{array}$$

The LSSVM prediction function is:(6)$$\begin{array}{c} f\left(x\right)=\overset{n}{\underset{i=1}{\sum }}\alpha_{i}K\left(x,x_{i}\right)+b \end{array}$$

By studying the LSSVM optimization model, it is known that the penalty coefficient r and the kernel parameter have a direct impact on the model’s precision. Therefore, selecting and optimizing r and α is crucial.

### 2.2. Sparrow Search Algorithm

The Sparrow Search Algorithm (SSA) is a 2020 meta-heuristic algorithm described by Xue et al. [17] that aims to do a local and global search by mimicking the foraging and anti-predator behavior of sparrows. Discoverers, entrants, and guards make up SSA. The discoverer has a high fitness value and provides the entrant with foraging direction and area; the entrant follows the discoverer to ensure predation efficiency; when the guard detects the predator, it immediately sends an alarm signal, and the entire population engages in anti-predator behavior or migrates to a safe area [22]. These are the specific steps [23].

The equation for updating the discoverer’s location is:(7)$$\begin{array}{c} x_{i,j}\left(t+1\right)=\left\{\begin{array}{c} x_{i,j}\left(t\right)\cdot \exp(-\frac{i}{\alpha T}),R<ST\\ x_{i,j}\left(t\right)+QL,\;\;\;\;\;\;\;\;\;\;\;\;R\geq ST \end{array}\right. \end{array}$$
where *t* is the current iteration number; T represents the maximum iteration number; α is a random number ([0, 1]); R represents the warning value of [0, 1]; ST represents the safety value. *Q* is a normal-distributed random number. When R<ST, there is no predator nearby and the finder can forage normally; when R≥ST, there is a predator nearby and all sparrows move to the safety area.

The formula for updating the entrant’s position is:(8)$$\begin{array}{c} x_{i,j}\left(t+1\right)=\left\{\begin{array}{c} Q\cdot \exp\left(\frac{x_{worst}\left(t\right)-x_{i.j}\left(t\right)}{i^{2}}\right),i>\frac{n}{2}\\ x_{p}\left(t+1\right)+\left|x_{i,j}\left(t\right)-x_{p}\left(t+1\right)\right|\cdot A^{+}L,i\leq \frac{n}{2} \end{array}\right. \end{array}$$

$x_{p}\left(t+1\right)$ represents the location in the global where the current adaptation is optimal; $x_{worst}\left(t\right)$ represents the location in the global where the worst adaptation is located; when $i>\frac{n}{2}$, the entrant moves closer to the discoverer in order to find a better location; when $i\leq \frac{n}{2}$, the entrant discards the current worse location. The column vector of a single sparrow at the same latitude is denoted by $A^{+}$.
(9)$$\begin{array}{c} A^{+}=A^{T}{\left(AA^{T}\right)}^{-1} \end{array}$$

The formula for updating the guard’s location is as follows:(10)$$\begin{array}{c} x_{i,j}\left(t+1\right)=\left\{\begin{array}{c} x_{best}\left(t\right)+\beta \cdot \left|x_{i,j}\left(t\right)-x_{best}\left(t+1\right)\right|,f_{i}>f_{g}\\ x_{i,j}\left(t\right)+K\left(\frac{\left|x_{i,j}\left(t\right)-x_{worst}\left(t\right)\right|}{\left(f_{i}-f_{w}\right)+\varepsilon }\right),f_{i}=f_{g} \end{array}\right. \end{array}$$

$x_{best}\left(t\right)$ is the global fitness optimal location; β is the step control parameter; K is a random parameter; $f_{i}$ is the current individual sparrow’s fitness value; $f_{g}$ is the global best fitness value; $f_{w}$ is the global worst fitness value; ε is a universal constant; When $f_{i}$ > $f_{g}$, the sparrow at the population’s edge is vulnerable to predator attack; when $f_{i}$ = $f_{g}$, the sparrow in the center of the population is in danger and has to be near other sparrows to lessen the likelihood of predation.

### 2.3. Optimized Sparrow Search Technique for Least Squares Support Vector Machines

The distribution of the initial population of a heuristic algorithm has a significant impact on its convergence speed and precision [24]. The standard sparrow search algorithm uses a random distribution of the beginning population in the search space. This approach has an unpredictably uneven distribution of population members, which impacts the accuracy of its predictions. The Monte Carlo method (QMC) fills a multidimensional hypercube cell with points as consistently as possible, making it economical and uniform for solving probabilistic issues. Among these, the Sobol sequence offers the advantages of a short computational cycle, rapid adoption, and greater efficiency when dealing with sequences of high latitude [25]. Using Sobol sequences, the initial population of the sparrow search method is optimized in this article. The detailed stages are listed in:

Let the optimal solution be $x^{n}\in \left[x_{min},x_{max}\right]$, and let $K_{n}$ be a random number in the range [0, 1]. The initial location of the population can then be described as [26,27]:(11)$$\begin{array}{c} x_{n}=x_{min}+K_{n}\left(x_{max}-x_{min}\right) \end{array}$$

The absence of effective control over the step length in the original sparrow search algorithm leads the program to readily converge on local optimal solutions. Consequently, nonlinear inertia weights are utilized to regulate the search range and convergence rate [28,29]. The weight of nonlinear inertia, w, is determined as:(12)$$\begin{array}{c} w=exp\left(1-\frac{t_{max}+t}{t_{max}-t}\right) \end{array}$$

The nonlinear inertia weights undergo adaptive adjustments to update the discoverer position as the number of iterations grows.
(13)$$\begin{array}{c} x_{i,j}\left(t+1\right)=\left\{\begin{array}{c} x_{i,j}^{t}\cdot w,\;R<ST\\ x_{i,j}^{t}+Q,\;R\geq ST \end{array}\right. \end{array}$$

Since the original method is incapable of solving the global optimal solution and quickly falls into the problem of the local optimal solution, the crisscross optimization algorithm is developed to optimize the sparrow search algorithm, thereby enhancing the model’s precision. The algorithm for crisscross optimization consists of horizontal crossover and vertical crossover. Horizontal crossover involves randomly pairing all of the fathers’ individuals in the population two by two in the same dimension to produce offspring individuals, and the offspring compete with the parents to select the most adaptable individuals to be retained [30]. Initially, the position of the guards is optimized and updated using the horizontal crossover algorithm. Assuming that the guard father individuals and that they are horizontally crossed in the *d*th dimension, the offspring are as follows [31,32]:(14)$$\begin{array}{c} MSx_{i,d}\left(t\right)=r_{1}\cdot x_{i,d}\left(t\right)+\left(1-r_{1}\right)\cdot x_{j,d}\left(t\right)+c_{1}\cdot \left[x_{i,d}\left(t\right)-x_{j,d}\left(t\right)\right] \end{array}$$
(15)$$\begin{array}{c} MSx_{j,d}\left(t\right)=r_{2}\cdot x_{j,d}\left(t\right)+\left(1-r_{2}\right)\cdot x_{i,d}\left(t\right)+c_{2}\cdot \left[x_{j,d}\left(t\right)-x_{i,d}\left(t\right)\right] \end{array}$$

$MSx_{i,d}\left(t\right)$ and $MSx_{j,d}\left(t\right)$ are the offspring individuals produced by lateral crossing from guards $x_{i,d}\left(t\right)$ and $x_{j,d}\left(t\right)$. $r_{1}$ and $r_{2}$ are random numbers between [0, 1], whereas $c_{1}$ and $c_{2}$ are random numbers between [−1, 1].

Guards generate offspring individuals in their own hypercube spaces and edges with a higher probability of horizontal crossing, boosting the algorithm’s global search capabilities.

Since the original algorithm lacks a special variational mechanism, it cannot effectively regulate individuals that have already reached a local optimum, thus preventing the search for the global optimum from continuing. Vertical crossover is an arithmetic crossover of all generated offspring individuals in different dimensions that increases the possibility that individuals who have fallen into the local optimum would escape it [32]. Therefore, it is necessary to optimize the original method using the crossed vertically. The following equation is constructed assuming that the offspring, $x_{i,d}\left(t\right)$, vertically cross over in dimensions $d_{1}$ and $d_{2}$.
(16)$$\begin{array}{c} MSx_{i,d}\left(t\right)=r\cdot x_{i,d_{1}}\left(t\right)+\left(1-r\right)x_{i,d_{2}}\left(t\right) \end{array}$$

$MSx_{i,d}\left(t\right)$ is the offspring produced by the vertical crossover of individual $x_{i,d}\left(t\right)$ in dimensions $d_{1}$ and $d_{2}$, whereas r is a random value between [0, 1].

Similar to horizontal crossover, all father individuals in a population are randomly paired two by two along the same dimension to produce offspring individuals, which then compete with their parents to retain the most adapted individuals. Individual sparrows can vary the population’s diversity and continuously improve the quality of the solution without losing critical data by employing this optimization technique.

Figure 1 shows the entire optimization process for LSSVM utilizing the ISSA algorithm.

### 2.4. Participants and Experimental Equipment

The literature method determined that human height and weight parameters have a significant effect on seat pressure distribution. To ensure the reasonableness and representativeness of this experiment, the height and body mass index (BMI) was used to screen the participants, and their body size had to fall between the 5th and 95th percentiles. According to the guidelines established by the World Health Organization, a BMI between 18.5 and 24.9 is regarded normal, less than 18.5 is considered thin, 25 to 29.9 is considered overweight, and 30 to 30.9 is considered obese [33]. Based on literature research [34], 30 valid participants were selected to participate in this experiment. Six were thin, ten were normal, eight were overweight, and six were overweight. This is shown in Table 1.

After counting the most prevalent seat surface materials on the market, three different seat surface materials, namely sponge, polyurethane foam, and latex, were chosen as samples. The material sample characteristics were kept consistent at 45 cm × 45 cm × 5 cm to control variables. Sponge cushions typically have a high density of 45 kg/cm^3^, resulting in exceptional resilience and support; polyurethane foam cushions, which are created by polymerizing isocyanate and hydroxy oxide, have excellent softness, elongation, and compression strength. The modulus of elasticity of latex cushions is between 2~4 MPa, with good high elasticity at room temperature, and as the production process typically involves foaming, the cushions indicate that they will develop dense, microscopic holes, resulting in excellent breathability.

Using a cushion pressure sensor manufactured by Tekscan in the United States, a body pressure measure system collects pressure data between the human body and the seat surface. The Tekscan cushion pressure sensor (shown Figure 2) is comprised of a polyester film consisting of multiple rows of ribbon conductors with a thickness of roughly 0.1 mm, whose outer surface is coated with a pressure-sensitive semiconductor compound. When an external force is applied to the sensing point, the resistance of the semiconductor varies in inverse proportion to the force. The pressure sensor utilized in this experiment consists of 32 × 32 detecting cells separated by 1.5 cm. Pressure cushion size 48 cm × 48 cm. 

Commonly utilized pressure distribution metrics are contact area, peak pressure, average pressure, force, etc [9,35]. Most experts and researchers believe that the best pressure map has a big contact area, a low peak pressure, an average pressure, and a high strength [13,36]. The contact area primarily represents the fit between the seat cushion and the human body. The softer the seat surface, the larger the contact area between the human body and the seat cushion. The average pressure reflects the support effect of the seat on the human body, which is primarily influenced by the human–chair interface shape, hardness, and other factors. The lower the average pressure, the more effectively the seat disperses the weight of the human body, and the less the human body feels the stimulation brought by the seat surface. Peak pressure and strength primarily define the seat surface’s stiffness. The stiffer the seat surface material, the higher the peak pressure and strength value [37].

The criteria for evaluating subjective comfort are based on the Likert scale [38]. The scale quantifies the subjective perceptions of the user in terms of scores, with points 1–7 corresponding to the intensity of sensory change, which corresponds to very uncomfortable, less comfortable, somewhat uncomfortable, average, somewhat comfortable, more comfortable, and very comfortable, respectively. The higher the score, the greater the level of comfort.

### 2.5. Experimental Producer

To decrease the influence of complicated environmental on the experimental results, the Student Innovation and Entrepreneurship Centre was chosen as the experimental site (Shenyang Aerospace University, Shenyang, China). The laboratory has ample space, a good light environment, and sufficient air temperature and humidity to offer a good external environment for the experimental area, which must be tested before the experiment; the results of these measurements are displayed in Table 2.

(1) To eliminate the subjective comfort bias produced by clothes, participants should wear long, loose-fitting cotton sweatpants with no items in their pockets. The pre-test was administered fifteen minutes before the experiment.

(2) The participants were instructed to maintain a stable sitting position in the allocated office chair for two minutes. To guarantee that the subject’s buttocks and thighs fit the seat surface appropriately, the participant’s tail hip bone was positioned at three-quarters of the seat surface while seated. The calf was perpendicular to the ground.

(3) On a seven-point Likert scale, participants assessed subjective comfort.

(4) After the conclusion of a single experiment, the participant rested for five minutes before beginning the next test, which consisted of the identical experimental steps (2) and (3) until the conclusion of the study.

(5) The software BPMS Research 7.0 installed on a laptop computer was used by the testers to collect real-time pressure distribution test data.

## 3. Model Development and Validation

### 3.1. Data Collection and Analysis

Each experimental data acquisition lasts 2 min, and the acquisition frequency of the experimental equipment is set to the default value of 5 frames per second. A single experiment contains 600 frames in total. When assessing the pressure data, the average value of each data set was chosen. The pressure data were opened in ASCII format and exported to an Excel spreadsheet. Before exporting the data, the experimental data must be pre-processed, first to remove extraneous pressure factors caused by the compression phenomenon between the seat cushion pressure sensor and the seat edge, and then to screen out certain abnormal values in the data to reduce the error of the experimental results. Kolmogorov–Smirnov normal distribution test was conducted on the collected pressure data and the subjective scale values, and it was found that some data did not obey the normal distribution, so the non-parametric test method was used for statistical analysis of the data.

Using the Friedman test, the means and standard deviations of the three sitting materials were compared. The contact area of the latex (461.32 cm^2^ ± 10.52 cm^2^) was the biggest, and the contact area of the sponge (303.62 cm^2^ ± 9.14 cm^2^) was the shortest, as indicated in Table 3. Latex had a much bigger contact area than the other two materials, and the contact areas of the three materials were statistically different at the 99% confidence interval (*p* < 0.001 **). The larger the contact area, the better the fit between the human buttocks and the seat surface, the softer the seat surface material, and the greater the human comfort. In terms of contact area, latex > polyurethane foam > sponge. A post-hoc pairwise comparison of the three-seat surface materials revealed that the contact area of each two materials was substantially different at the 99% confidence interval (*p* < 0.001 **), as shown in Table 4.

The peak pressure value of the sponge is the highest (246.05 mmHg ± 9.21 mmHg), while the peak pressure value of latex is the lowest (221.13 mmHg ± 11.15 mmHg). The peak pressure values of the three materials differ significantly within the 99% confidence interval. The peak pressure represents the softness of the seat surface material and the force on the hip joint. The higher the peak pressure, the less soft the seat surface material. The greater the force on the hip joint, the greater the likelihood that the human body is uncomfortable. Table results indicate that, in terms of peak pressure, latex > polyurethane foam > sponge, that is, polyurethane foam seat surface material softness is the worst, and the corresponding human hip joint force is the greatest. As shown in Table 4, a post-hoc pairwise comparison of the three seat surface materials revealed that, at the 95% confidence interval, the peak pressure of each of the two materials was substantially different (*p* < 0.001 **).

Mean pressure of the sponge (101.65 mmHg ± 1.77 mmHg) was the highest and the mean pressure of the latex (90.69 mmHg ± 1.02 mmHg) was the lowest, and there was a significant difference in the mean pressure of the three-seat materials at the 95% confidence interval. The average pressure is the arithmetic mean of the total pressure at all measurement points. The average pressure reflects the stiffness of the seat surface and the overall force on the human hip. The average pressure is also influenced by the shape of the seat surface. From the point of view of average pressure, it can be concluded that sponge > latex > polyurethane foam. As shown in Table 4, a post-hoc two-by-two comparison of the three-seat surface materials showed a significant difference in mean pressure for each of the two materials at the 99% confidence interval (*p* < 0.001 **).

The force and peak pressure response trends were essentially identical. The force value of the sponge (149.32 mmHg ± 5.41 mmHg) is also the largest, while the force value of latex (83.64 mmHg ± 2.99 mmHg) is the smallest; the force value of latex is significantly greater than the force values of the other two office seats, and the force values of the three-seat surface materials differ significantly at the 99% confidence interval (*p* < 0.001 **). Force reflects the rationality of the seat surface material distribution. The greater the value of force, the poorer the rationality of the seat surface material distribution. The greater the human pressure stimulation induction, the less in line with human physiology design characteristics. Latex is superior to polyurethane foam and sponge in terms of force. As demonstrated in Table 4, with a post-hoc pairwise comparison of the three seat surface materials, the force values of each pair of materials differ significantly at the 99% confidence interval (*p* < 0.005 **).

In terms of subjective comfort, latex was rated higher than the other two materials, with the sponge receiving the lowest rating. At the 99% confidence range, there was a significant difference between the three-seat materials (*p* < 0.001 **). Regarding subjective comfort, latex is superior to polyurethane foam and sponge. A post-hoc pairwise comparison of the three-seat surface materials revealed substantial differences in subjective comfort for each of the two materials, as indicated in Table 4.

The correlation study between body pressure parameter data and subjective comfort was performed using Spearman correlation analysis, as shown in Table 5. The correlation coefficients between contact area and subjective comfort were all larger than 0.8, indicating a strong positive correlation. The correlation coefficients between peak pressure, average pressure, force, and subjective comfort degree were all less than −0.8, indicating a strong inverse relationship. The greater the contact area, the greater the subjective comfort, and the greater the values of peak pressure, average pressure, and force, the worse the subjective comfort.

### 3.2. Establish the SSA-LSSVM Prediction Model

This research uses the mean absolute error (MAE), root mean square error (RMSE), and goodness of fit (R^2^) as evaluation indicators to quantify the accuracy of the prediction model. The input layer is comprised of six indicators, including Body Mass index (BMI), gender, and pressure distribution indicators, while the output layer is the subjective comfort score of the user [39]. The normalized data were inserted into the ISSA-LSSVM model for training, and r = 187, α = 10 were determined to be the ideal parameters for building the prediction model. Figure 3 shows the prediction results of the ISSA-LSSVM. Table 6 compares the predicted values obtained from this prediction model to the true values.

The prediction results show that the goodness of fit R^2^ is 0.92, the root mean square error RMSE is 0.29, and the real value and the predicted value fit well. The average absolute error MAE is 0.24, and the model has good predictive performance. The ISSA-LSSVM approach can solve the problem of large prediction deviation in small sample prediction by using an improved sparrow search algorithm to optimize the necessary parameters of LSSVM, and the prediction accuracy can reach 95.75%. The relative standard deviation (RSD) of the model was calculated to be 5.99%, indicating that the model has good stability.

### 3.3. Model Validation

Validation was conducted using the PSO-LSSVM (Particle swarm optimization for optimizing least squares support vector machines), GWO-LSSVM (Grey Wolf optimization for optimizing least squares support vector machines), and SSA-LSSVM algorithms, respectively, to confirm that the prediction model of ISSA-LSSVM is more stable and accurate.

As can be seen from Figure 4, PSO has the least number of iterations and the fastest iteration speed, but has a larger fitness value, meaning that the search is the worst; SSA and GWO have comparable and relatively good fitness values, but the iteration speed of SSA is significantly faster than GWO; ISSA has a faster iteration speed while ensuring that the fitness is small enough, indicating that ISSA has a faster speed to jump out of the local optimum and find the global optimum solution, and has a better global search capability.

From Figure 5 and Table 7, it can be seen that the prediction performance and fitting effect of PSO-LSSVM are relatively poor, and the model is the most unstable with the highest RSD of 14%. SSA-LSSVM already has better prediction performance and fitting effect, but the model is generally stable with a higher RSD of 11.23%. After optimizing the SSA algorithm, the ISSA-LSSVM model outperformed the SSA-lssvm model in terms of prediction accuracy and model stability, with R2 improved by 0.2, RMSE reduced by 0.03, MAE improved by 0.05, RSD reduced by 5.41%, and prediction accuracy improved by 1.83%. This illustrates the value of the multi-strategy SSA algorithm and the superiority of ISSA-LSSVM.

## 4. Conclusions

There is a tight relationship between office chairs and modern people’s lifestyles. A comfortable office chair can enhance users’ experience. As the carrier between the user and the seats, the material of the seat surface directly affects the user’s comfort. Therefore, it is vital to investigate and assess the comfort of the office chair’s seat material. Through the body pressure distribution test experiment, objective measurement data were obtained. Combined with the user’s subjective comfort collaborative analysis, a high correlation between the user’s subjective comfort and objective measurement data was demonstrated. Finally, the user’s BMI index, gender, contact area, peak pressure, average pressure, force, and other objective measurement data were used as input variables, and using the user’s subjective comfort as the output variable, a sparrow search algorithm was developed to optimize the least squares support vector machine seat surface material comfort prediction model. With a prediction accuracy of 95.75 percent, the model accurately predicts the comfort of the office chair’s seat surface material. It was discovered that the model has a greater prediction accuracy than PSO-LSSVM, GWO-LSSVM, SSA-LSSVM, and other prediction models, indicating that the ISSA-LSSVM model possesses superior global search capability and stability.

## Figures and Tables

**Figure 1 sensors-22-09822-f001:**
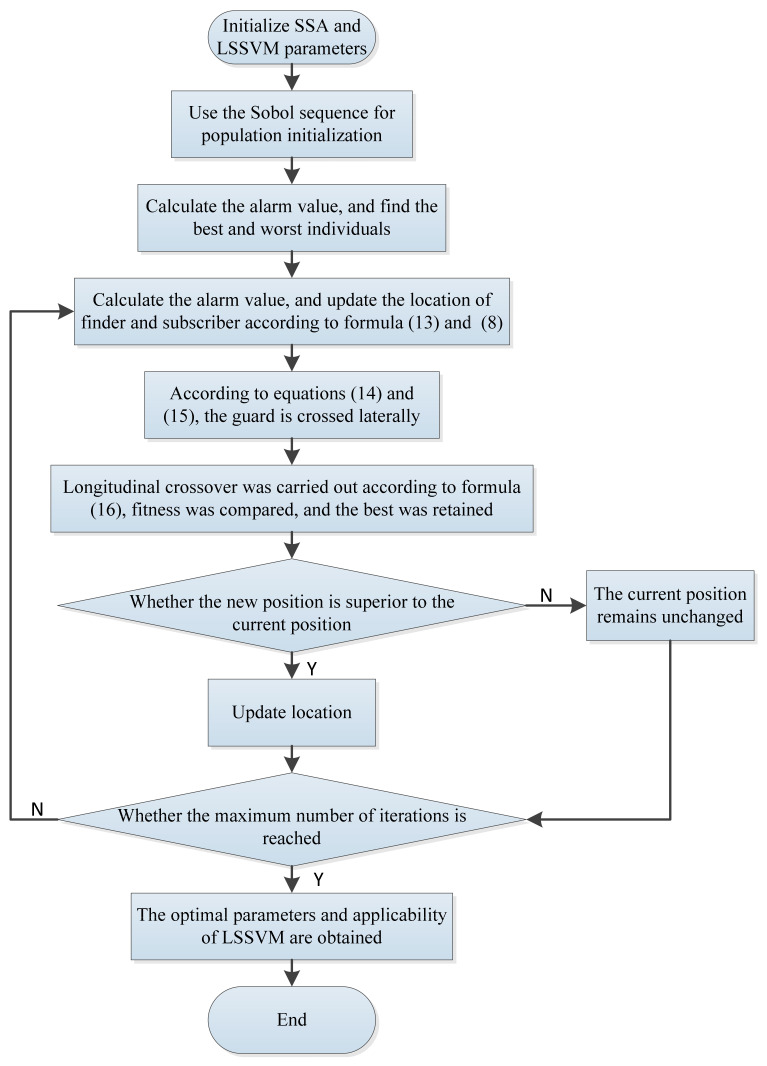
The flow chart of the ISSA-LSSVM.

**Figure 2 sensors-22-09822-f002:**
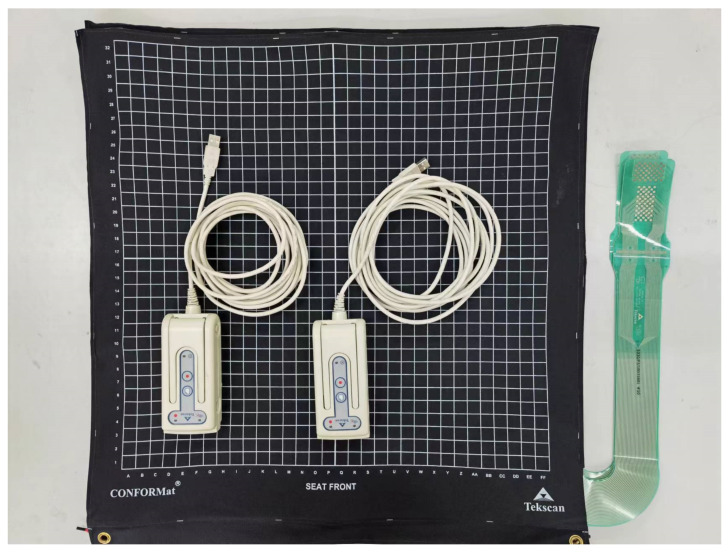
Body pressure distribution measurement system.

**Figure 3 sensors-22-09822-f003:**
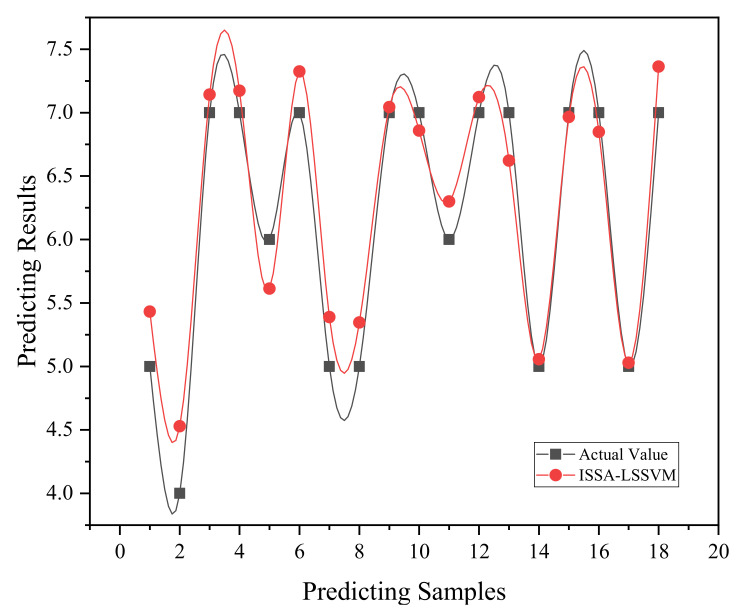
ISSA-LSSVM model prediction results.

**Figure 4 sensors-22-09822-f004:**
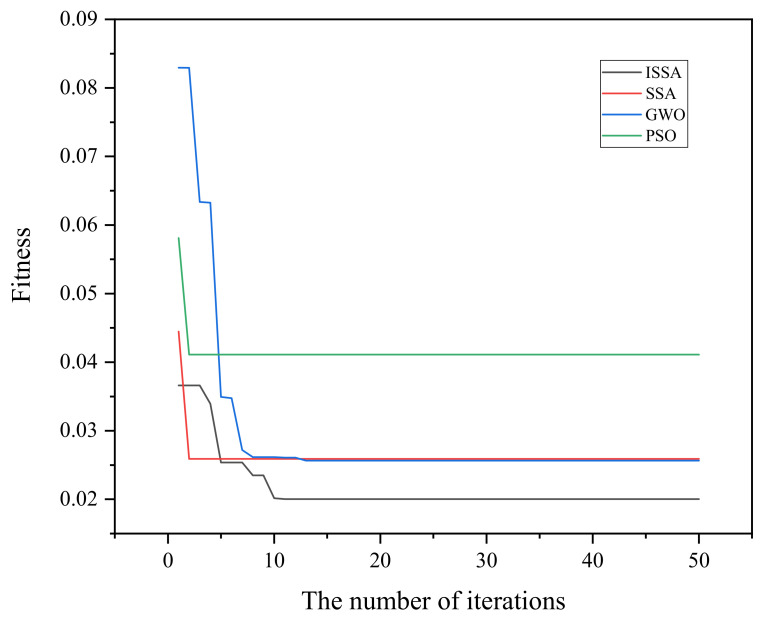
Iteration process of each optimization algorithm.

**Figure 5 sensors-22-09822-f005:**
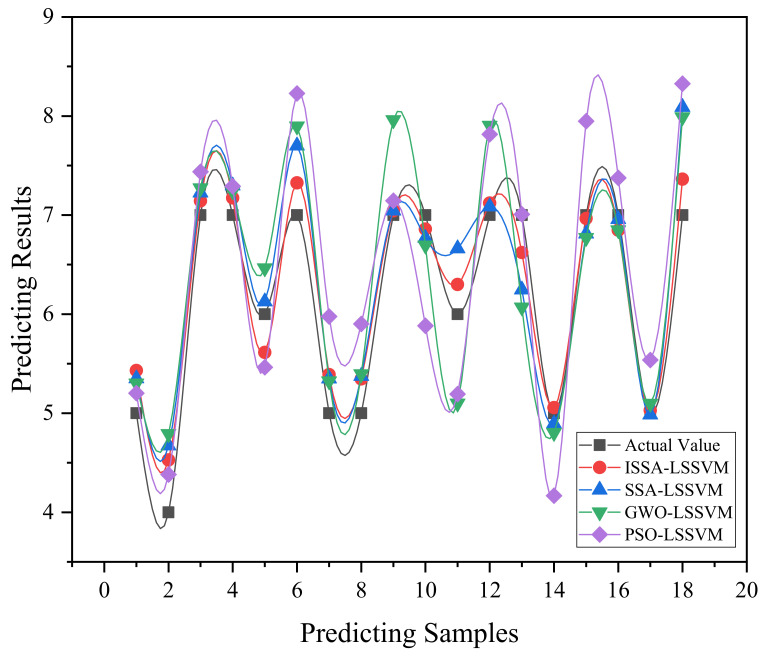
Comparison of prediction results.

**Table 1 sensors-22-09822-t001:** Basic information about the subject.

	Number	Age	Height (cm)	Weight (kg)
Male	15	31.22 ± 10.73	174.33 ± 6.6	73.37 ± 7.79
Female	15	30.38 ± 12.26	165.25 ± 3.9	56.25 ± 11.59
Sum	30	30.82 ± 11.48	170.06 ± 7.2	65.31 ± 13.28

**Table 2 sensors-22-09822-t002:** Environment of the experimental area.

Environmental Index	Test Value	Comfort Value
Indoor temperature (°C)	17 ± 0.4	15–18
Relative humidity (%)	47% ± 5%	40–60%
Wind speed (m/s)	0.33 ± 0.1	0.3

**Table 3 sensors-22-09822-t003:** The effect of seat surface material on body parameters.

Parameter	Seat Surface Material Types			Chi-Square Value	Significance b (*p*)
	PUF	Sponge	Latex		
Contact Area (cm^2^)	375.33 ± 6.55	303.62 ± 9.14	461.32 ± 10.52	1200	<0.001 **
Peak Pressure (mmHg)	237.94 ± 6.36	246.05 ± 9.21	221.13 ± 11.15	926.688	<0.001 **
Average Pressure (mmHg)	90.69 ± 1.02	101.65 ± 1.77	93.98 ± 0.86	1197.506	<0.001 **
Force (mmHg)	105.68 ± 2.52	149.32 ± 5.41	83.64 ± 2.99	1200	<0.001 **
Subjective Comfort	5.05 ± 1.13	4.23 ± 1.11	6.14 ± 1.08	33.342	<0.001 **

Note: ** indicates a significant difference at the 99% confidence interval.

**Table 4 sensors-22-09822-t004:** After pairwise evaluation of seat surface material groups.

Parameter	Seat Surface Material Type					
	PUF-Sponge		Sponge-Latex		PUF-Latex	
	Z	Sig	Z	Sig	Z	Sig
Contact Area (cm^2^)	−21.222	<0.001 **	−21.222	<0.001 **	−21.222	<0.001 **
Peak Pressure (mmHg)	−17.072	<0.001 **	−21.195	<0.001 **	−20.445	<0.001 **
Average Pressure (mmHg)	−21.222	<0.001 **	−21.222	<0.001 **	−21.206	<0.001 **
Force (mmHg)	−21.222	<0.001 **	−21.222	<0.001 **	−21.222	<0.001 **
Subjective Comfort	−3.286	<0.001 **	−4.080	<0.001 **	−3.551	0.037 **

Note:”**” indicates a significant difference at the 99% confidence interval.

**Table 5 sensors-22-09822-t005:** Correlation between subjective comfort and body pressure parameters.

Parameter	Correlation Coefficient
	Subjective Comfort
Contact Area (cm^2^)	0.958
Peak Pressure (mmHg)	−0.921
Average Pressure (mmHg)	−1
Force (mmHg)	−1

**Table 6 sensors-22-09822-t006:** The predicted value is compared with the actual value.

Sample	Predicted Value	True Value	Relative Error
1	5.4320	5	0.0864
2	4.5292	4	0.1323
3	7.1426	7	0.0204
4	7.1725	7	0.0246
5	5.6130	6	0.0645
…			
18	7.3629	7	0.0518

**Table 7 sensors-22-09822-t007:** ISSA-LSSVM is compared with other prediction models.

Prediction Model	R^2^	RMSE	MAE	RSD	Prediction Accuracy
ISSA-LSSVM	0.92	0.29	0.24	5.99%	95.75%
SSA-LSSVM	0.84	0.39	0.30	11.23%	93.73%
GWO-LSSVM	0.77	0.49	0.48	10.91%	91.51%
PSO-LSSVM	0.67	0.58	0.44	14.67%	89.09%

## Data Availability

Not applicable.

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
