# Peer review of "Comfort Prediction of Office Chair Surface Material Based on the ISSA-LSSVM"

_sensors, 2022, doi:10.3390/s22249822_

Round 1
Reviewer 1 Report
In this paper, the authors used several models to predict the comfort of office chair surface materials. This paper provided some interesting information. However, some problems need to be solved before it could be considered a research paper.
1.To evaluate the prediction ability, the root mean square error (RMSE) and goodness of fitting (R2) cannot be used as criteria. These criteria are to be used for evaluating the fitting-ability of models. All experimental data should be split into two data sets. The first data set is used for fitting-agreement of the model evaluation and the other data set is used for predicting evaluation.
2.The age distribution in Table 1 show these samples are too homogeneous. The 22 samples are young persons, maybe all graduate students. The samples cannot represent normal human beings. More subjects including more ranges of age and weights should be recruited to perform the test.
3. Equation (17) is used to calculate the sample numbers and ? is the allowable error in this equation. How do you know the allowable error in your experiment is the same as the literature Kolich[34].
4. The criteria for evaluating subjective comfort are based on the Likert scale. The scale is not continuous data and is not a normal distribution. How can you perform the statistical test (ANOVA) without the normal test?
5. The quality of Figures 5-8 must improve.
6. The concepts of the statistical test are not correct. Please ask a statistician to check the statistical method and rewrote this paper.
Author Response
Thank you for your criticism and correction, the comments you gave me were very valuable. I have revised the manuscript in accordance with your suggestions. Your comments are a valuable asset to me and I thank you again.
For details, please see the attachment.

Reviewer 2 Report
Reviewer’s report:
Title: Comfort prediction of office chair surface material based on the ISSA-LSSVM.
In this work, authors reported the development of a robust and accurate model for predicting the surface comfort of office chairs by using sparrow search algorithm that has been optimized which called as an improved sparrow search algorithm. The idea of manuscript was good with enough discussion which may beneficially for readers who work in the field. However, it still needs revision or modification in parts as shown in the comments below.
1. At line 104., the value of “1” in Eq.1 is right?., At line 249, typo: “i”?., What's mean of stars “**”, it needs an explanation.
2. At lines 106 and 110, Don’t use the same character of “b” with different meaning!
3. At Eq.10, what is “Ô‘”?, At line 336, what is BMI index?, At line 351, what is PSO and GWO?, You need to mention it at least once at first.
4. At line 227, It will beneficially if you add the properties of the sponge, polyurethane foam, and latex.
5. Fig.2 is a pressure map? You need to make a discussion for Fig.2 in relation to the mapping of the pressure. Add the dimension, pressure unit, or scale bar at the Figure.
6. Fig. 3 is an experimental flowchart?, You have to revise it!
7. At lines 304-305, Which table?, based on the Table 3, in fact sponge > polyurethane foam > latex. The sentence needs a modification to be clear.
8. At Figs. 4,5,6,7,8, “X” and “Y” axes lines are fuzzy, the legends are too small, the graph lines are also too thin and fuzzy, grid lines are unclear.
9. The authors claim that the ISSA-LSSVM model possesses superior global search capability and stability, why?
Author Response

(The authors gave the same response as above.)

Reviewer 3 Report
General Comment: This study predicts the selection of comfortable office chair surface materials and improves their comfort in using office chairs using human pressure distribution data from 22 participants and the user’s subjective comfort evaluation methods.
Specific Comments:
1. The quality of all the figures is poor and should be improved. English should be improved by further proofreading the paper and removing the typos and grammatical errors from the text. There are also some complex sentences that do not read well.
The introduction section needs to be updated using more recent references. Authors should enrich the introduction section.
10.1016/j.apergo.2014.12.010
10.3390/en12193695
10.1016/j.applthermaleng.2015.08.077
10.1016/j.buildenv.2018.05.061
2. The abstract doesn't meet the journal standard, and it must be completely revised. It should briefly focus on these points :
Purpose: Clearly define the purpose and importance of your research. This includes a statement of the problem or issue.
Methodology: State the research methods used to answer your question.
Results: Summarize the main research results.
Conclusion: What are the implications of your research?
3. Please eliminate multiple references such as "Gradually, artificial intelligence technology has been used in the subject of ergonomics [11–13]." After that, please check the manuscript thoroughly and eliminate ALL the lumps in the manuscript. This should be done by characterizing each reference individually and mentioning 1–2 phrases per reference to show how it is different from the others and why it deserves mentioning. Multiple references are useless to a reader and can even be considered plagiarism if the authors use them without thoroughly studying the references that they are citing. In this case, each reference should be justified as to why it is used, and at the very least a brief assessment should be provided.
4. Too many non-content words may indicate wordiness. Consider rewriting to eliminate the following words: the, for, of, into, via, as well as, their, with, are, also, and which. For the text's clarity, the authors should refrain from using additional words, as most of them are meaningless filler words or some archaic words, e.g., "respectively", "thus", "hence", "therefore", "furthermore", "thereby", "basically", "meanwhile", " wherein", "herein", "hitherto", "Nonetheless", "Perceivably" , "whereas",etc. ?
Please closely follow the ISO symbol presentations:
For example: - 3 (/m3) in superscript; 20 mm, which must be 20 mm.
Please carefully check the complete manuscript, including the pictures.
5. Why are the effects of gender not considered in this study? Because this is a prediction study, the effects of both genders must be considered and included.
10.1016/j.buildenv.2019.01.058
10.1016/j.buildenv.2019.01.007
Author Response
Thank you for your criticism and correction, the comments you gave me were very valuable. I have revised the manuscript in accordance with your suggestions. Your comments are a valuable asset to me and I thank you again.
For detials, please see the attachment.

Round 2
Reviewer 1 Report
The content of the revised version has been improved significantly. All problems were replied appropriately
Reviewer 3 Report
The authors have incorporated all the suggestions in the revised manuscript.